# Challenges Affecting Access to Health and Social Care Resources and Time Management among Parents of Children with Rett Syndrome: A Qualitative Case Study

**DOI:** 10.3390/ijerph17124466

**Published:** 2020-06-21

**Authors:** Javier Güeita-Rodriguez, Pilar Famoso-Pérez, Jaime Salom-Moreno, Pilar Carrasco-Garrido, Jorge Pérez-Corrales, Domingo Palacios-Ceña

**Affiliations:** 1Department of Physical Therapy, Occupational Therapy, Physical Medicine and Rehabilitation, Research Group of Humanities and Qualitative Research in Health Science of Universidad Rey Juan Carlos (Hum&QRinHS), Avenida Atenas s/n, Alcorcón, 28922 Madrid, Spain; javier.gueita@urjc.es (J.G.-R.); domingo.palacios@urjc.es (D.P.-C.); 2Department of Nursing, Servicio Madrileño de Salud, 28004 Madrid, Spain; pilarfamoso50@gmail.com; 3Department of Physiotherapy, Universidad Francisco Vitoria, 28223 Madrid, Spain; jaime.salom@ufv.es; 4Department of Medical Specialities and Public Health, Universidad Rey Juan Carlos, Alcorcón, 28922 Madrid, Spain; pilar.carrasco@urjc.es

**Keywords:** health services accessibility, parents, rare disease, Rett syndrome, qualitative research

## Abstract

Rare diseases face serious sustainability challenges regarding the distribution of resources geared at health and social needs. Our aim was to describe the barriers experienced by parents of children with Rett Syndrome for accessing care resources. A qualitative case study was conducted among 31 parents of children with Rett syndrome. Data were collected through in-depth interviews, focus groups, researchers’ field notes and parents’ personal documents. A thematic analysis was performed and the Standards for Reporting Qualitative Research (SRQR) guidelines were followed. Three main themes emerged from the data: (a) essential health resources; (b) bureaucracy and social care; and (c) time management constraints. Parents have difficulties accessing appropriate health services for their children. Administrative obstacles exist for accessing public health services, forcing parents to bear the financial cost of specialized care. Time is an essential factor, which conditions the organization of activities for the entire family. Qualitative research offers insight into how parents of children with Rett syndrome experience access to resources and may help improve understanding of how Rett syndrome impacts the lives of both the children and their parents.

## 1. Introduction

Rett syndrome (RS) is a rare genetic neurological condition, that affects about 1:10,000–15,000 of female infants [1]. This syndrome is characterized by gait abnormalities, acquired microcephaly, repetitive hand movements, loss of speech, and breathing disorders [1,2]. These functional problems may be accompanied by other comorbidities such as gastrointestinal disorders, epilepsy, and scoliosis [3,4]. RS presents a high complexity of symptoms, causing children to experience social, physical and communication challenges, which may restrict their social participation and activities of daily living [3]. Access to adequate care in many rare disease patients is limited and presents barriers for access to treatment options, together with educational and/or social opportunities [5]. Many patients with rare diseases experience barriers in accessing medical care, and less than 10% receive disease-specific treatment [6]. Furthermore, rare diseases constitute a major economic burden, regardless of a country’s size and demographics [7].

In addition, rare diseases also have an impact on the economic, psychosocial and physical well-being of families [8]. Overall, families have limited information for making decisions regarding the treatment of the disease and the relief of symptoms [9,10]. The uncertainty of having a rare disease, together with delays in diagnosis and lack of knowledge about care and treatment needs [11,12], also impacts access to services and disease management [13]. Families undergo a sort of “pilgrimage” as they must go back and forth between institutions and health specialists due to the absence of structured and targeted programs and policies in the management of rare diseases [14].

The distribution of health expenditure varies among countries, dependent on factors such as the burden of diseases and the priorities of each health system [15,16]. In the case of rare diseases, due to their low prevalence, high variability and numerous subtypes, a more global and comprehensive approach is required on behalf of various public and private organizations aimed at preventing significant morbidity and premature mortality, and improving the quality of life and socioeconomic resources of affected people [17,18,19].

Living with a rare disease implies facing social situations such as stigma and isolation, the search for normality and the need for support [20]. Previous authors [21] describe how support from the patient’s community is fundamental to drive programs and make legislative changes to improve care for rare diseases. Research on the experience of having a rare disease indicates that care and service needs are met by a combination of poor quality of care and barriers to accessing services [11,22]. In a review of the unmet needs of parents of children with rare diseases, social, informational and emotional needs were identified as the most pressing [23]. The authors also suggested the need for further research in this area to better address the unique care needs of these families. The experience and individual perspective of the parents concerned is relevant, just as life experiences, ambitions and emotional needs should be considered when treating children with RS [3,24]. It is necessary to explore certain questions: what are parents’ experiences regarding their ability to access appropriate care for their children with RS? What is their day-to-day experience with healthcare, and the social care system? This study examines the experiences of parents of children with RS in terms of the challenges related to access to care resources.

## 2. Materials and Methods

This study followed the guidelines for conducting qualitative studies established by the Standards for Reporting Qualitative Research (SRQR) [25]. Qualitative research methods provide a further understanding of the beliefs, values, and motivations related to individual health behaviors [26].

### 2.1. Design

A qualitative descriptive case study design was used [27,28]. Qualitative methods are indicated to help understand the beliefs, values, and motivations that underlie individual health behaviors [29]. This type of case study is useful for describing a phenomenon or intervention and the real-life context in which it took place [27]. The study topic was the impact of RS disease on parents, in terms of access to health and social care resources, as well as considering time management constraints.

### 2.2. Research Team

Prior to beginning the study, the positioning of the researchers [30] was established during two briefing sessions, considering the theoretical framework, their beliefs, and their motivations for this research [26,29], as displayed in Table 1.

Six researchers (four men and two women) participated in this study, including an occupational therapist (JPC), a pharmacologist (PCG), two physical therapists (JGR, JSM), a Registered Nurse (PFP) and a research nurse (DPC). Three of the participants (DPC, JPC, JGR) were experienced in qualitative research, were not involved in clinical activity, and had no prior relation with the study participants.

### 2.3. Context

Rett syndrome is a type of rare disease [14]. Rare diseases only affect a small proportion of the population, between 40 and 50 cases per 100,000 people [14]. Families receive important support from family associations who provide true and up-to-date information, collaborating in care for the children and funding research [31]. The participating parents were recruited from the Mi Princesa Rett Association (https://miprincesarett.es/) and the Spanish Rett Syndrome Association (http://www.rett.es/).

### 2.4. Participants

The inclusion criteria were: (a) parents of children who were diagnosed with RS, and/or their legal guardians; (b) a diagnosis of RS by a pediatrician and/or a neurologist; (c) any variation of RS; and (d) informed consent signed by parents. The exclusion criteria consisted of: (a) a diagnosis of RS, not confirmed by the pediatrician and/or neurologist; and (b) failure to sign the informed consent.

### 2.5. Sampling Strategies

Purposive, critical-case sequential sampling was used, based on relevance to the research question, resulting in the recruitment of 20 participants [29,32]. Snowball sampling was also incorporated, whereby individuals selected for the study assist researchers in identifying other potential participants [29,32]. This led to the recruitment of a further 11 participants.

Data collection and sampling was pursued until data saturation was achieved after the inclusion of 31 parents, at which point no new information emerged from the data analysis [26,29].

### 2.6. Recruitment

The directors of both participating associations were responsible for introducing the parents to the researchers. Subsequently, in an initial face-to-face meeting, researchers explained the study’s purpose and design to all those who fulfilled the inclusion criteria. Participants were then given a one-week period to allow them to decide whether or not they wished to participate. Thereafter, in a second face-to-face session, written informed consent and permission to record the interviews was provided. All the initially selected parents agreed to participate in the study.

### 2.7. Data Collection

Data collection took place between April and October 2016. This study sought to act as an in-depth multi-perspective holistic enquiry regarding the study phenomena, requiring multiple data collection tools [26]. No previous theoretical or conceptual model was used prior to data collection. The initial stage of data collection involved semi-structured, in-depth interviews using a question guide (Table 2), to gather information on specific topics of interest [26,29]. During these interviews, the researchers noted the key words and topics identified in the parents’ responses and answered further questions when further clarifications were needed [26]. This enabled researchers to obtain relevant information from the parents’ perspective. The question guide was developed based on a prior literature review and the researchers’ experience.

During the second stage, a focus group (FG) was conducted using a question guide (Table 2), in order to examine different perspectives within the same group, acquire understanding of the problems faced by the group and identify values and norms [29,32,33]. The FGs were led by a moderator, based on a uniform structure. The question guide was sufficiently focused to gather information on the area of study, although sufficiently open to stimulate interaction and debates among participants [29,32,33].

Both the interviews and FGs were audio-recorded and transcribed verbatim. Overall, 19 interviews were conducted, and two FGs, with seven and five participants, respectively. In total, 1333 min of data collection were recorded, with 1073 min corresponding to the first stage and 260 min corresponding to the second stage. During the first stage, interviews lasted between 73 and 183 min, whereas in the second stage, the FG lasted between 96 and 164 min. The interviews took place at the associations or at the parent’s home, as preferred by each parent. In addition, 21 researchers’ field notes and one personal letter were collected.

### 2.8. Data Analysis

A thematic, inductive analysis was performed [29,32,34]. This method is congruent with case study designs [26,27]. Verbatim transcriptions were made for each of the FGs, in-depth interviews, researchers’ field notes, and for the documents provided by participant’s [29,32,34]. The thematic analysis process [29,32,34] consisted of identifying the most descriptive content to convert the data into meaningful units, and thereafter reduce and identify the most common meaningful groups. Thus, clusters of meaningful units were generated, i.e., similar points or content, enabling topics to emerge describing the experiences of the study participants [29,32,34]. (See Appendix A). This procedure was used separately for the interviews, FGs, and personal documents. Subsequently, joint meetings were conducted to pool the results of the analysis and to discuss the data collection and analysis procedures. In the event of differences in opinion, theme identification was performed based on a consensus among the research team members. Finally, the research team held joint meetings to present, combine, integrate and identify the final themes [29,34]. No data analysis software was used.

### 2.9. Rigor

The Standards for Reporting Qualitative Research (SRQR) [25] (http://www.equator-network.org/) were followed. Trustworthiness of the data was established following the criteria by Guba and Lincoln (Table 3) by reviewing issues concerning data credibility, transferability, dependability, and confirmability [26,29,30]. These methods of increasing rigor are often used in case-study designs [35].

### 2.10. Ethical Considerations

This study complied with the Declaration of Helsinki, and ethical approval was obtained by the Rey Juan Carlos University Clinical Research Ethics Committee (code: 220220161516). All subjects provided their informed consent prior to participation.

## 3. Results

In total, 31 participants (17 women, 14 men) were included, with a mean age of 45.38 (SD ± 10.85) years. Overall, 83.9% (*n* = 26) were married, 6.4% (*n* = 2), were widows/widowers, and 9.7% (*n* = 3) were separated. Nineteen families were interviewed, including 19 children with RS, as no family had more than one member with RS among them. Regarding the gender of the affected children, 14 were female and 5 were male. The mean age of children with RS was 12.57 (SD ± 9.02) years and the mean age at diagnosis was 4.50 (SD ± 3.56) years. (See Appendix A—Demographic and clinical features of the family participants.) Three main themes emerged from participants’ narratives: (a) Essential health resources, with four subthemes; (b) Bureaucracy and social care, with two subthemes; and (c) Time management constraints, with two subthemes. (See Table 4, Themes that emerged from participant narratives.)

Below, we include some of the patients’ narratives taken directly from the interviews, focus groups and personal letters in relation to the three emerging themes and subthemes.

### 3.1. Theme 1. Essential Health Resources

This theme describes the experiences of parents in relation to identifying the essential health services, such as the genetic diagnosis, and when they sought professional help at the level of primary care or hospital care. Parents outlined the difficulties encountered and their expectations surrounding health care and health professionals.

#### 3.1.1. Genetic Diagnosis

Final confirmation of the diagnosis of the disease requires costly genetic testing. Access to genetic diagnosis and coverage of its cost by the Spanish public health system differs across regions in Spain. Not all parents have access to this test, in which case, the parents must bear the cost.

“*I have paid for my son’s genetic testing. It wasn’t covered on Social Security where we live… I needed to know exactly what was wrong with him. I have to spend the money. They’ve opened a genetics clinic in Valencia*.”(P15)

Moreover, parents related how the protocols for applying genetic testing give preference to girls over boys, as it is a female-dominated disease, which may delay the diagnosis for some children.

“*It took too long to do it to my son, they put us on hold. We later learned that they didn’t do it because he was a boy*.”(P16)

#### 3.1.2. Primary Care

Parents described the lack of specific programs for the early detection of rare diseases, and more specifically RS.

“*I’ve been telling the doctor that there has been something wrong with my daughter since, since her first birthday, she doesn’t walk, she doesn’t talk, she doesn’t manipulate things, she doesn’t do anything, I took her to the pediatrician and he said to me ‘the problem is that you don’t stimulate her. Don’t worry, until she is eighteen months old you can’t activate specific protocols. We lost a year and a half*.”(P31)

Due to the lack of knowledge regarding the disease, parents reported that the check-ups carried out in primary care were unsuitable, making them feel that they knew more about what was happening to their child than the doctors themselves.

“*I went in and he said to me, ‘put her there and I’ll measure her,’ and I said, ‘she can’t stand up’, he measured her, weighed her and that’s it! In the end you come out of there saying, ‘I’ll take matters into my own hands…’ Well, they don’t tell you anything else, I think it’s astonishing… If they don’t know about the disease, at least try to help those parents… And in the end, that’s what you become, a doctor*.”(P1)

Some parents acknowledged that when they made observations about their child’s condition in relation to the presence of new signs, or queries about follow-ups, they perceived a lack of understanding regarding what it is like to have a child with RS. This translates into a feeling of not receiving proper care from the professionals.

“*I kept insisting to the pediatrician that it wasn’t normal for my child to fall and not be able to get up. He said that it was normal because she was delayed in her development and was very fat… but I could see that something was wrong with her and the doctors were giving me the cold shoulder*.”(P20)

#### 3.1.3. Specialized Hospital Care

The participants expressed that where one lives determines access to adequate hospital care. Thus, parents living in large cities (e.g., Madrid or Barcelona) have ample access to services and resources, while in other smaller cities such resources are lacking. The main consequence is that, in the event of requiring specialized hospital care, parents and children with RS must travel long distances to other hospitals away from their homes, having to cover all the related costs (transfer, maintenance, apartment rental, etc.).

“*I go to Madrid… From the very beginning, I was referred to Madrid, and what great expenses I must cover in order to live there, without any aids, again! The thing is, based on the few things they have done to her here, no, no, I don’t want it, they are not prepared, these children are lacking so much specific care*.”(P15)

One of the more special cases is when parents have wanted a second medical opinion on their child’s treatment or the disease’s evolution. Under the Spanish national health system, this is a right that is protected by law, the cost of which should be covered by the public health system. However, many parents were unable to access this option and had to cover these costs themselves.

“*I spent two months there going every day, sitting in front of the office, until they gave me the letter of referral to see another doctor… That’s something meaning that if you ask for a second opinion, you can go anywhere in Spain… But hey! If there is a law, then it’s important to comply with it, or else you’ll have to pay for it again*.”(P28)

The parents described how, at the hospital, the children’s follow-up visits were overcrowded and over-saturated, the room was noisy, and space was cramped, generating a lot of stress for both the children and the parents.

“*After two hours of waiting he said that he couldn’t examine her, the doctor himself said, ‘the thing is, I have to see thirty children…’ He can’t explore her because she was stressed, because of her stiffness… Because of how long she’s been waiting, she became upset and screamed, the place where we have to wait is hopeless*.”(P20)

#### 3.1.4. Treatment and Care

Parents spoke of how there is no specific treatment for RS, and therefore the therapeutic program is very extensive and variable.

“*With a child who you must take to multiple therapies, like others, who take them to after-school activities, in her case, she needs those therapies to live, what we never know is which of these is the most appropriate*.”(P1)

Likewise, treatments may vary according to region where they live, and parents have difficulties in early access to medical treatments.

“*So, searching for therapies, we couldn’t find where to go, neither did we receive calls from the ones that existed, time went by… so the doctor taught us to do the exercises at home and we had to do them ourselves until she could start the sessions*.”(P15)

Families also faced difficulties in accessing public health care in the home for their daughters (nursing, ventilatory support, etc.), and with the lack of public resources which they could access. Extensive differences were noted between the different Spanish regions.

“*With oxygen 24 h a day. We’ve had it for six months… On oxygen for three years… With nebulizers and oxygen… We still have it all there, because it wasn’t easy to take care of at home, nobody taught us how to manage things at the beginning. However, for other people from the association who live in Madrid, everything is easy, with a nurse who goes to the home*.”(P13)

### 3.2. Theme 2. Bureaucracy and Social Care

This topic described how parents present difficulties in accessing public social aids aimed at acquiring special equipment (adapted beds), renting or modifying infrastructure (modifications and home adaptations), acquiring orthopedic materials (orthoses and prostheses), and/or buying/renting cars and accessing assistive resources such as early intervention care for girls with RS, and other assistive resources (day centers, etc.).

#### 3.2.1. Fighting Bureaucracy

Parents have to deal with extensive bureaucracy and legal technicalities that they are not used to, in order to apply for social benefits. This was a recurrent theme among all the parents interviewed:

“*I have tried to fight, and I have always lost with the Administration because I am at a loss with the bureaucracy, this is what has to change… And the fact is that families with people who are greatly affected do not have to endure what we are going through. All the endless bureaucracies that exist for any necessary procedure. Incalculable timeframes*.”(P24)

Parents mentioned how facing bureaucracy is “a continuous struggle” to enforce their children’s recognized rights to receiving care, feeling forgotten, even, at times, mistreated by public institutions in this regard.

“*The family members who have a serious pathology within the family, make us lose heart… Because it is a never-ending struggle, with the issue of the Dependency Law… For us, it’s a great help when it was put into practice… But of course, you have to go to the Social Worker and pester them until you manage to activate it. They make you feel lonely, helpless*.”(P24)

One of the aspects of this struggle is the absence of clear, understandable and accessible public information on the available aids. The advice on how to process subsidies and manage the bureaucracy often comes from other parents who have undergone the same situation. As a result, parents’ and patients’ associations are deemed essential.

“*Matters regarding having a dependent child and the issue of disability, this information was given to us at the Early Intervention Centre. There we asked for the first steps, however, access to public resources is not easy, so, many of the things you ask and request is because other families from the association that we go to tell you about it, as they have already fought with the administration before*.”(P4)

Another aspect of this ongoing struggle takes place upon applying for aids. Parents are confronted by the official administration employees. As an entity, these are perceived as another obstacle, as these civil servants often add further obstacles and hindrances when handling applications.

“*I requested for a new report for the therapies several times, but the official didn’t know how to tell me that my daughter’s documentation was inside a paperwork bundle in Merida… The administration had taken away an assistant - the official who processed the reports - and she didn’t have time to manage so much paperwork… So, the problem was there, and everything was stalled… You become outraged*!”(P19)

In short, the bureaucracy and processing of aids is a great difficulty for all parents and is experienced as a great problem, which makes them feel defenseless.

“*The most frustrating thing is that any help that is proposed costs money, and you know that it is going to cost you almost more effort and time to fight it… very much*.”(P21)

#### 3.2.2. Privatization of Social Care

The parents perceived a change before and after the economic crisis that hit Spain between 2008 and 2012. They explained that many public aids have either been reduced or discontinued (aids for assistance at home, for purchasing equipment and orthopedic appliances, etc.):

“*During the years of the crisis, many of our equipment expenses did not go through the Social Security, and some of them started to go through in the year 2012. So, there are many families who, as of 2012, have stopped paying for tests or certain orthopedic things*.”(P10)

Faced with this situation, many private entities have appeared to replace the services provided by the public system, albeit charging for their services. However, many families are unable to afford these services.

“*This has an impact on our financial resources, because the effort I had to make, to try to give my daughter the best possible options, was to take her to one of the private centers that cover the lack of public services…and now I don’t work*.”(P3)

An example of the rise of private entities or clinics is the access to early intervention centers. These centers provide treatment and care that prevent or improve the disability associated with RS. Early intervention centers are mostly private entities that charge for their services. However, not all parents can access them.

“*Treatments that are not covered come out of our pockets… because you need a physical therapist and a speech therapist every day. All of this treatment is individual and private at the early intervention centers*.”(P4)

This generates feelings of frustration in parents, since a care resource exists that will help their daughters, albeit they cannot access it, and there is no public aid for these services.

“*It was frustrating to realize that he needed therapy every day and we couldn’t give it to him… We sacrificed everything until we couldn’t pay for it and we had to reduce it to fewer times per week because there is no help… until now he that he no longer receives it*.”(P27)

Although home care is very expensive, parents stressed their desire to keep their daughters at home as long as possible and be responsible for their care. Institutionalization at a long-term care facility was not an option for them, even though being at home entails greater financial and personal costs.

“*And when they turn 21, perhaps they are taken to a day care center. And unfortunately, my daughter needs the therapies from age 4 to 70. We don’t want to leave her at the center all day, we prefer to keep her at home and take her to her therapies, even though it’s very difficult for us financially*.”(P1)

The use of permanent homes is reserved for cases where parents are unable to care for their daughters. However, the problem is that when that time comes, many parents cannot afford the cost.

“*I was thinking about putting her in a nursing home when she was older, when we are no longer here. But when it got really bad, we couldn’t find anything in the public system, and the private ones were untouchable*.”(P25)

### 3.3. Theme 4. Time Management Constraints

Time management was a relevant issue for many parents, as time constraints are related to many instances during treatment and care processes. Examples of such issues include the early detection of the first symptoms, a definite genetic diagnosis in order to begin treatment, delay in receiving public social care, early access to early care intervention to prevent sequelae and disability, time spent on travelling to other cities in search of medical help, etc. Time management can mean whether your children receive adequate care or not.

“*So, what kind of life do I lead? What you see from the outside, is that my life is about going at full speed through life and not having a minute’s rest*.”(P15)

*“For me, the paramount thing is about organization, to be able to organize my time*.”(P4)

“*You try to make do as best as you can… it’s non-stop, time is a necessary resource, like money, but, oh well, it’s what we have to live with*.”(P9)

#### 3.3.1. Immediate Time

Parents stressed the need for a distribution of their time in the “immediate” term. They must distribute their time on a daily basis to be able to cover the daily needs of the whole family, taking into account that one of their members (the daughter with RS), presents a major disability and dependency.

“*Maybe what I need is more time, if I had more time, I would do many things, but of course I need time to work and earn money to live, I need time to be with my children… And I need time to overcome my daughter’s serious illness, these things I try to combine with as much balance as possible and it can become very complicated*.”(P24)

One of the aspects parents have learned is that time becomes a “present” experience. The moment, the now, is the key, and the future is uncertain:

“*I’m worried about now, I’m not worried about tomorrow, I’m worried about now… I’ve learned to live in the ‘today’, I haven’t learned to live in the ‘tomorrow’*.”(P15)

In addition, they must dedicate time to fulfilling their own individual and social roles, such as going to work, university, etc.

“*We used to be better organized. But now that she has gotten worse, we have had to reorganize outings to save time. When my husband works, I don’t move. And the other way around, I take advantage of it to go to class and study when he’s free*…”(P6)

#### 3.3.2. Scheduled Time

For parents, the distribution of family time and organizing the family dynamics always depends on the care and treatment required by the child with RS.

“*We plan day by day, we can’t consider anything else… no plans for Easter, let’s see if this weekend, if she’s OK, we’ll go, the idea of arranging trips in advance or saying we’ll go in the summer, in August, etc… Nope, you stop doing that, because I don’t know how my daughter will be or what she’ll need*.”(P3)

At home, they cannot make projections. At any time, their daughters can have a seizure and they have learned not to plan activities, to avoid unnecessary frustration:

“*As you don’t know when the crisis is coming, how can you organize your life? You just can’t! Nowadays everybody is organized… But this really hits you… You have to live life minute by minute, so you don’t become overwhelmed*.”(P19)

A necessary aspect of time scheduling is traveling to the medical centers to receive the treatment and care they need. Some families must spend much of their time traveling to other cities or towns.

“*I was always on a tight schedule with her… I was going to take her to the doctor and then to the physical therapist in the next town, but then of course I got stuck in traffic… I would never get there on time and something would always happen to me… I would always get there stressed out*.”(P6)

Finally, the time spent on all the bureaucracy and administrative procedures to apply for grants, visit public centers, including waiting times in public services, etc., is worth noting. The time spent on all these procedures was considered as “lost time” or a “waste of time”. Thus, time is an essential resource for parents, and when nothing is achieved, or nothing about their situation is fixed, they perceive these procedures as a poor investment of their time:

“*The disability assessment center is running late. They called us a year after it had expired, having complained many times on the phone without getting anywhere, until I got in touch with the director of the assessment center, after several failed trips to see him…they are part of all the work hours I lose, nobody values the time we waste*.”(P24)

## 4. Discussion

The results of our study emphasize that access to the necessary resources in both health and social care is perceived as a “continuous struggle”, which translates into a waste of time and money for parents. This struggle includes the search for a diagnosis, encountering barriers to primary care and specialized care (hospitalization), and finding health professionals who do not know the disease and the needs of their children or their own needs. In addition, they encounter great difficulties in accessing social support, mainly due to administrative procedures and bureaucracy. The solution that parents usually find is to bear the financial cost themselves if they want to access the benefits earlier or wait and feel that they are losing a lot of time to obtain these from the public sector.

Our participants described barriers to seeking and receiving a diagnosis, and subsequently to accessing appropriate treatment. The information needs of parents were not adequately addressed by primary care physicians or specialists. The authors believe that the lack of protocols on the diagnosis of RS and the gap in knowledge about rare diseases may be the cause of these situations. The participants’ accounts revealed difficult experiences, specifically related to the lack of information on the diagnosis process, barriers to access certain services and poor coordination of care. Our results coincide with previous studies on the challenges faced by families with rare diseases navigating the healthcare system [36], in order to access adequate and effective care [11,12,37], and the economic and social consequences that RS specifically has on families by them assuming all costs [38,39].

Due to the lack of up-to-date, accurate, proven and accessible information on RS, many participants adopted and sought a proactive attitude and behavior for managing their daughter’s disease. Thus, many parents acknowledged “feeling like doctors”. This behavior and active involvement of parents often appears in other rare diseases [40]. Von der Lippet et al. (2017), in their systematic review on living with a rare disease, identified the notion of the “expert patient” [41]. This “expertise” also appears in parents, when the patient is a minor, and the parents become “expert caregivers” [36]. Parents tend to remain active in the management of their child’s health [42]. However, this expertise, developed by the patients themselves and their families, is rejected by health professionals, or they have difficulty accepting the experience of those affected regarding their own process [43]. This contributes to poor care for people with RS in our findings. The authors of this study believe that promoting a more equal relationship between health professionals and parents is essential, since parents can provide up-to-date knowledge of the effect of treatments on their children, detect changes, and identify new signs and symptoms, while constantly caring for their children.

Out-of-pocket payments and extensive travel in search of resources were common among participants in this study. Along these lines, Bauschbaum et al. [36] show how parents with children with rare diseases assume most of the expenses related to the disease, throughout their children’s life. In addition, parents of children with RS have to modify their working conditions (changes in working hours, shorter working days) in order to care for their children, and/or attend medical appointments, which may worsen their financial situation due to a reduction in their income [44].

Despite the multitude of health professionals and medical services involved, there is no cohesive and coordinated framework for focused care of RS. This is hampered by the variability of symptoms and the diagnostic difficulty that characterizes RS [1,2]. According to our findings, this situation explains parents’ perception of lack of information and poor coordination within the health care system. This uncertainty is experienced with discomfort by parents, due to the lack of clear indications regarding the disease and the multiple therapies and interdisciplinary interventions that their children require (physical therapy, occupational therapy, drugs, etc.). Several authors stress the importance of avoiding leaving the task of coordinating care and assistance in the hands of the family, with all the emotional and psychological burden that this entails [36,45].

Our findings highlight difficulties in accessing some public social benefits. These results show how parents fight against bureaucracy, and do not perceive themselves as potential users of the public administration support system. Caring for a child with a rare disease is extremely difficult as families feel helpless in their “continuous struggle”. Cardinalli et al. [20] describe how parents of children with rare diseases must fight to obtain adequate care and the consequent rights needed to access the various social support services and to overcome bureaucratic difficulties. Other authors, point out how people with rare diseases and their families are perceived as invisible, experiencing a sense of disconnection and silence from the health care providers [45,46]. In addition, previous studies [23,36,47] describe how parents have difficulties making sense of the system and “navigating” through the different levels of health care and disability support providers in order to seek help from the administrations.

Other key aspects are the search for support among peers or among parents experiencing the same situation with their children affected by RS or other rare diseases. Affected associations provide parents with information, and help parents in their decision making and purchase of aids and resources to care for their children [36]. They also help parents feel less isolated, by being able to share experiences [20,48].

The families in our study did not have access to social services and financial subsidies to cover their expenses. The coverage of economic costs together with the aid provided by public administration facilitates care and coverage of family needs, enabling the ability to cater for the unique requirements and difficulties of each family [49,50]. Health care and social support models are organized for temporary care for health problems, and are not prepared to provide coverage for chronic, disabling and complex diseases such as RS or other rare diseases [12,41]. Our results show how parents took their children to private specialized care services, such as early intervention centers. Health and social care providers need to collaborate with families to develop appropriate health and social care policies for rare diseases [51]. These results should help medical and social services to provide multidisciplinary support with specific models of care for RS, supporting the involvement and active participation of those affected and their families [50].

In the present study, time is perceived as an asset and a valuable resource in the life of the parents of children with RS. To the best of our knowledge, no previous studies have identified time as being an essential resource for parents with children with RS. Indirectly, the study by Spillman et al. [52] shows how time delay in diagnosis, and therefore in possible treatment, affects parents of children with rare diseases. Other authors [11,12], have shown how the time spent in receiving public social assistance, early access to early care to prevent sequelae and disability, and time lost when travelling to other cities in search of medical help leads to great differences in whether or not the children receive adequate care.

Regarding the practical implications of our study, these findings reflect a practical demand for a greater coordination between the health and social sectors to establish programs that help parents with children with RS and complex chronic diseases [53]. In addition, training programs must be developed for health professionals, in order to update knowledge of RS and other rare diseases. However, health and social managers need to be made more aware of the pressing need to create specific support programs for children with RS and their families, because of the great physical, functional, psychological and social impact of the disease. Finally, coordination between primary care and specialist care is crucial to avoid the pilgrimage of parents across different levels of care in search of help.

Among the strengths of our study, this is the first research of this type describing the perspective of parents with children with RS and their difficulties in managing resources in Spain. Also, the use of different tools for collecting qualitative data has allowed us to broaden and deepen the perspective of parents.

The limitations of our study include the fact that our results cannot be extrapolated to all parents who have children with RS, due to the design used. In addition, children were excluded from the interviews due to their limited communication and cognitive skills. Nonetheless, these results may help professionals understand parents’ experience, and their difficulties in accessing resources and aids for caring for their children. Finally, it was not possible to include both partners in all cases, as in some cases one partner refused to participate in the study. In addition, no physician participated as a member of the research team. However, the authors believe that the interdisciplinary research team that participated in the study is made up of a wide variety of health professionals, thus avoiding the predominance of one professional perspective over another.

## 5. Conclusions

This study describes the perceived difficulty among the parents of children with RS for gaining access to health and social care resources, and the management of time constraints. Our findings shed light on how families navigate the health and social systems, with important implications for clinical practice.

These data may be useful in assisting with the development of programs and policies that address the needs of parents with children with RS and/or reduce the difficulties and/or challenges they encounter. Understanding parents’ experiences can help foster more equitable and closer care of families. Our study provides a basis for further studies that address the impact of RS on parents’ lives, on their quality of life, and on the quality of health care received.

## Figures and Tables

**Table 1 ijerph-17-04466-t001:** The positioning of the researchers.

Theoretical framework	The researcher’s approach was based on a constructivist paradigm. This was built on the assumption that human beings construct their own social reality, and understanding is built via increasingly nuanced reconstructions of individual or group experiences.
Beliefs	Rett Syndrome (RS) typically becomes apparent during the first year of life, presenting as a neurodevelopmental delay. This experience can be particularly traumatic, affecting family dynamics. It is necessary to examine the parents’ perspective regarding relevant aspects that impact their daily life.
Motivation for the research	To understand the parents’ experiences and how they manage health and social resources for their children suffering from RS. The limited availability of international qualitative research on this subject and the lack of research conducted in Spain warrants the need for qualitative research examining the parents’ perspective.

**Table 2 ijerph-17-04466-t002:** Questions guide.

Investigated Theme	Questions
Experience with the illness	What is your experience and perspective of Rett syndrome?
Management of health resources	How do you manage your child’s symptoms? What care needs does your child have? How do you access health services? What barriers and/or facilitators have you encountered? What aspect was most relevant for you?
Management of social resources	What social needs does your child have? How do you access social services and community support? What barriers and/or facilitators have you encountered? What aspect was most relevant for you?
Impact on the family	How did the illness influence your family life and the relationship with the family members? What aspect was most relevant for you? What is your everyday life like?

**Table 3 ijerph-17-04466-t003:** Trustworthiness criteria.

Criteria	Techniques Performed and Application Procedures
Credibility	Investigator triangulation: the interviews were analyzed by three researchers. During subsequent team meetings the analyses were compared, to identify categories. Triangulation of data collection methods: focus groups and interviews (unstructured and semi-structured) were conducted and researcher field notes were kept. Participant validation: participants were asked to verify the data obtained during data collection and analysis. All participants were offered the opportunity to review the audio or written records as well as the subsequent analysis, to confirm how the researchers interpreted their experience.
Transferability	In-depth descriptions of the study procedures, detailing the characteristics of researchers, participants, contexts, sampling strategies, and the procedures for data collection and analysis.
Dependability	Audit by an external researcher who reviewed the research protocol, with a special emphasis on aspects concerning the methods applied and study design. In addition, an external researcher verified the description of the coding tree, the major themes, verbatim quotations, coding of verbatims, and the description of themes.
Confirmability	Investigator triangulation, participant triangulation, and data collection triangulation. Researcher reflexivity was carried out via the performance of reflexive reports and by describing the rationale for the study.

**Table 4 ijerph-17-04466-t004:** Themes that emerged from participant narratives.

Themes	Subthemes
3.1. Essential health resources	3.1.1. The genetic diagnosis 3.1.2. The primary care 3.1.3. Specialized hospital care 3.1.4. Treatment and care
3.2. Bureaucracy and social care	3.2.1. Fighting bureaucracy 3.2.2. Privatization of social care
3.3. Time management constraints	3.3.1. Immediate time 3.3.2. Scheduled time

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
