# Peer review of "Challenges Affecting Access to Health and Social Care Resources and Time Management among Parents of Children with Rett Syndrome: A Qualitative Case Study"

_ijerph, 2020, doi:10.3390/ijerph17124466_

Round 1
Reviewer 1 Report
The article deals with problems that affect people with rare diseases and their families. They were presented in qualitative terms. Unfortunately, they show that despite many actions taken, not only patients from Spain feel that they are still objectified and they are not the subjects. This is particularly evident in areas where the competences of the healthcare system and social welfare decussate. If the research protocol allows, the following would be advisable to supplement: 1. Information, e.g. on the duration of the disease and other sociodemographic data of the respondents, not just marital status. 2. justification of the composition of the research team, especially in the context of the profession - there is no doctor, and some comments refer to the activity of this professional group. 3. Information about an external researcher - profession. 4. The applications require rewording, even to the form of practical demands, so that they are not only a summary of the results obtained.Author Response
Please see the attachment.

Reviewer 2 Report
Thank you for this manuscript.
This study sought to explore the experiences of a group of parents of children with RS regarding the challenges affecting the access to care resources. This is an important research question. More research is needed in this subject.
The choice of a qualitative study design is correct.
The structure of the manuscript is good, and it is easy to follow how the authors have conducted the study.
I think it is important knowledge, and that this manuscript is needed.
The data includes interviews from 31 parents, and parents to the same child probably have similar experiences. It is unclear how many children that is represented in the data collection. Please describe if there exists more than one parent from each family. It was 17 women and 14 men included in this study. Was it 31 families, or were some of these participants belonging to the same family?
Two of the themes are similar to the questions in the interview guide, e.g.
Question: Management of health resources - Theme: Access to health resources
Question: Management of social resources - Theme: Access to social resources.
A qualitative analysis can be either inductive or deductive. An inductive analysis gives themes that is based on the parents´ narratives. More descriptive analysis often describes what the parents tell. In contrast, a deductive analysis is based on predetermined categories. In the results, the authors describe themes that are similar to the research questions, and this seems to be a deductive analysis. The third theme, however, seems to be inductive. The authors have described that they did an inductive analysis. However, if this is done I think that the authors need to rewritten the results, and the themes need to show an inductive analysis. Please, describe a result that is based on an inductive analysis.
Either did I not understand, or did you not conduct an inductive analysis. Therefore, please explain clearly why themes and questions have the same content.
The writing thematic, inductive analysis. In my opinion, an inductive analysis need to get themes that emerge from the text. The analysis need to answer the question "what". For example, I like the theme "Time management constraints". This theme explain the situation for the parents.
Round 2
Reviewer 2 Report
Thank you for this revised version of the manuscript.
I really appreciate the supplementary file that explain the relation between the children with Rett Syndrome and the parents who participated in the study. I also think that the information about the children gave a deeper understanding about the families that were included in this study.
I appreciate your clarifications of the analysis process, and your description of your interpretation of inductive and deductive analysis. The combination of these clarifications and the revised titles of the themes strengthen the trustworthiness of the results. I think that the revised titles of the themes make it easier for the reader to understand that your research questions not were predetermined categories. I also like that you describe in the method that your analysis not was driven by a predetermined model or theory. I think that the supplementary file of the analysis process is necessary to have for transparency in the analysis process.
I think that this manuscript now is suitable for publication.